# Morphological, Transcriptome, and Hormone Analysis of Dwarfism in Tetraploids of *Populus alba* × *P. glandulosa*

**DOI:** 10.3390/ijms23179762

**Published:** 2022-08-28

**Authors:** Yongyu Ren, Shuwen Zhang, Tingting Xu, Xiangyang Kang

**Affiliations:** 1National Engineering Research Center of Tree Breeding and Ecological Remediation, College of Biological Sciences and Technology, Beijing Forestry University, Beijing 100083, China; 2Key Laboratory of Genetics and Breeding in Forest Trees and Ornamental Plants, Ministry of Education, Beijing Forestry University, Beijing 100083, China; 3Beijing Laboratory of Urban and Rural Ecological Environment, Beijing Forestry University, Beijing 100083, China

**Keywords:** tetraploid, plant hormone, transcriptome, co-expression, poplar 84K

## Abstract

Breeding for dwarfism is an important approach to improve lodging resistance. Here, we performed comparative analysis of the phenotype, transcriptome, and hormone contents between diploids and tetraploids of poplar 84K (*Populus alba* × *P. glandulosa*). Compared with diploids, the indole-3-acetic acid (IAA) and gibberellin (GA_3_) contents were increased, whereas the jasmonic acid (JA) and abscisic acid (ABA) contents were decreased in tetraploids. RNA-sequencing revealed that differentially expressed genes (DEGs) in leaves of tetraploids were mainly involved in plant hormone pathways. Most DEGs associated with IAA and GA promotion of plant growth and development were downregulated, whereas most DEGs associated with ABA and JA promotion of plant senescence were upregulated. Weighted gene co-expression network analysis indicated that certain transcription factors may be involved in the regulation of genes involved in plant hormone pathways. Thus, the altered expression of some genes in the plant hormone pathways may lead to a reduction in IAA and GA contents, as well as an elevation in ABA and JA contents, resulting in the dwarfing of tetraploids. The results show that polyploidization is a complex biological process affected by multiple plant hormone signals, and it provides a foundation for further exploration of the mechanism of tetraploids dwarfing in forest trees.

## 1. Introduction

Polyploids have more than two sets of complete chromosomes and occur widely in nature [1,2]. In recent years, increasing attention has been paid to the study of plant polyploidy. Many previous studies have shown that polyploidization can lead to changes in plant growth, morphology, physiology, and biochemistry [3,4]. Some polyploid plants hold a significant advantage in vegetative growth compared with diploids. For example, triploid poplar [(*Populus pseudosimonii* × *P. nigra* ‘Zheyin3#’) × (*Populus × beijingensis*)] [5], *Eucommia* triploids [6], *Eucalyptus urophylla* triploids [7], and *Liriodendron* tetraploids (*Liriodendron × sinoamericanum*) [8] all show the characteristics of larger leaves, faster growth, and taller plants. However, not all polyploids show superior growth. For instance, tetraploid *Eucalyptus urophylla* [9], autotetraploid apple (*Malus* × *domestica*) [10], tetraploid *Populus hopeiensis* [11], hexaploid *Populus* (*Populus tomentosa × P. bolleana*) *×* (*P. alba × P. glandulosa*) [12], octaploid *Populus hopeiensis* [13], and tetraploid *Lycium ruthenicum* [14] grow slowly and show plant dwarfing. Thus, the role of polyploidization as a creative evolutionary force that produces novelty can be modulated by its effect on phenotypes [15].

Recent studies have extensively examined the effects of polyploidization on plant development at the transcriptome level. With a change in ploidy, gene expression may be modified significantly, resulting in changes in phenotypic characteristics [16]. For instance, overexpression of *GROWTH-REGULATING FACTOR 5* (*GRF5*) in triploid poplar enhances the expression of genes in the cytokinin signaling pathway, as well as promotes leaf growth and expansion [17]. Compared with diploids, autotetraploid energy willow *Salix viminalis* exhibited significantly higher levels of gibberellin, cytokinin, and jasmonic acid, resulting in thicker stems and larger leaves [18]. Diameter at breast height and leaf size of autotetraploid birch (*Betula platyphylla*) are greatly increased, and the phenotypic changes are mainly associated with upregulation of auxin (*AUXIN/INDOLEACETIC ACID* (*AUX/IAA*) and *GRETCHEN HAGEN 3* [*GH3*]) and ethylene (*ETHYLENE RESPONSE FACTOR 1/2* (*ERF1/2*)) signal transduction-related genes [19]. The significant increase in leaf area and plant height of tetraploid mulberry (*Morus alba*) is due to downregulation of ARABIDOPSIS RESPONSE REGULATOR (ARR) and upregulation of the gibberellin receptor GA-INSENSITIVE DWARF 1 (GID1), which alter the contents of cytokinins, gibberellins, and ethylene [20]. In addition, the dwarf phenotype of autotetraploid apple might be due to downregulation of *AUXIN RESPONSE FACTOR* (*ARF*) genes, which leads to partial interruption of the auxin and brassinolide signal transduction pathways, thus resulting in reduced contents of auxin and brassinolide [10]. The elevated expression of *ARF* and *MYB* genes in tetraploid *Populus* ((*Populus pseudosimonii* × *P. nigra* ‘Zheyin3#’) × (*Populus × beijingensis*)) decreases the contents of gibberellins and auxin, as well as increases the content of abscisic acid (ABA), thus accelerating chloroplast degradation and leaf senescence [21]. The dwarfing of tetraploid Chinese cabbage (*Brassica rapa* subsp. *pekinensis*) is caused by the downregulation of crucial genes involved in the regulation of brassinolide synthesis, resulting in inhibition of brassinolide synthesis [22]. These studies suggest that leaf development and plant dwarfing are the result of the interaction of several hormones.

*Populus alba* × *P. glandulosa* clone 84K has gradually become an important model material for research in forest molecular biology because of its high rates of differentiation and genetic transformation in vitro [23]. With the completion of whole-genome sequencing of poplar 84K, the clone is ideal for examination of the mechanisms of polyploid trait variation among perennial trees [24]. Therefore, in this study, diploid (2*n* = 2*x* = 38) and tetraploid (2*n* = 4*x* = 96) individuals of poplar 84K were used as study materials. Transcriptome sequencing analysis, combined with phenotypic indicators and hormone contents, and weighted gene co-expression network analysis (WGCNA) were used to clarify the regulatory relationship between transcription factors (TFs) and hormone signaling pathway genes and to analyze the effect of differential expression of hormone signaling pathway genes on the slow growth of tetraploids. The results provide a basis for further exploration of the dwarfing mechanism in tetraploids and provide a theoretical foundation for inhibition of plant growth to generate and utilize dwarf forms of trees.

## 2. Results

### 2.1. Comparative Analysis of Phenotypic and Anatomical Characteristics between Tetraploids and Diploids

To understand the growth patterns of diploid and tetraploid poplar 84K at different stages of development, the plant height at 1, 3, and 5 months after transplantation was measured. Compared with the diploids, the tetraploids grew slowly and showed dwarfism. The average height of the tetraploids was 60.73 ± 3.65 cm at 3 months, whereas the average height of the diploids was 87.30 ± 3.94 cm (Appendix A). The height of the tetraploids was approximately 30.44% lower than that of the diploids (Figure 1A,B).

To observe anatomical differences between the diploids and tetraploids, paraffin-embedded sections of stem internodes and leaves were examined. In general, in longitudinal sections of the stem internodes, the cells of the tetraploids were shorter and wider than those of the diploids to some extent, whereas the cell area of the leaf in transverse section was larger in the tetraploids. The average length and width of the stem internode cells of diploids were 70.47 ± 5.20 μm and 26.54 ± 4.18 μm, whereas those of the tetraploids stem internode cells were 54.82 ± 11.61 μm and 45.63 ± 4.94 μm, respectively (Figure 1C). The cell area of the main veins of diploid leaves was 715.19 ± 88.37 μm^2^, whereas that of tetraploid leaves was 2588.37 ± 462.70 μm^2^ (Figure 1D; Appendix A).

### 2.2. Transcriptome Sequencing Data Analysis

To analyze the gene expression differences between triploid and diploid leaves of poplar 84K during continuous growth, development, and senescence, the first, fifth, and 11th leaves (numbered from the shoot tip) were selected as experimental materials, which were labeled as D1, D5, D11, T1, T5, and T11, respectively. Three biological replicates were analyzed for each leaf node sampled, and the genome sequence for poplar 84K was selected as the reference genome. The transcriptome of the 18 leaves was compared and analyzed. A total of 137 G of valid data were obtained from the 18 cDNA libraries. The Q20 and Q30 quality scores attained at least 98.38%, and the overall GC content was more than 44% (Appendix A). On the basis of the comparison of the sequencing reads and reference genome, more than 80.37% mapped reads could be found in all libraries (Appendix A). These results showed that the selection of the reference genome was appropriate and the sequencing results were acceptable; thus, they were suitable for subsequent analysis.

Using fold change (FC) ≥ 2 or FC ≤ 0.5 (log_2_FC ≥ 1) and *p*-value < 0.05 as the screening threshold, 7917 differentially expressed genes (DEGs) were identified in the leaves at the three nodes. In total, 1622 DEGs were detected in the first leaf, comprising 1080 upregulated genes (66.6%) and 542 downregulated genes (33.4%), 3747 DEGs were detected in the fifth leaf, consisting of 1840 upregulated genes (49.1%) and 1907 downregulated genes (50.9%), and 2548 DEGs were identified in the 11th leaf, including 1417 upregulated genes (55.6%) and 1131 downregulated genes (44.4%) (Figure 2A). A Venn diagram showed that 447 DEGs (7.3%) were differentially expressed between diploids and tetraploids at all three leaf nodes (Figure 2B). A cluster analysis heatmap clearly divided the 18 samples into two groups based on the expression level of DEGs, which separated the diploids and tetraploids (Figure 2C). These results indicated that the biological replicates of each sample were highly similar, and that the ploidy and leaf positions may lead to differences in gene expression patterns.

### 2.3. Functional Enrichment Analysis of DEGs in the Transcriptomes

To explore the functions of the DEGs, GO annotation and KEGG pathway enrichment analysis were performed on the DEGs of tetraploids and diploids. All annotated DEGs were classified into three functional categories: biological process, cellular component, and molecular function. The most highly enriched GO terms are shown in Figure 3A–C (*p* < 0.05). Among the 20 most enriched GO terms in the biological process category, 2–187 DEGs in the first leaf were involved in ‘hormone transport’, ‘sucrose synthesis’, and ‘response to auxin, brassinolide, JA, and other hormones’ (Figure 3D). In the fifth leaf, 4–267 DEGs were involved in the ‘signaling pathways of cytokinin, gibberellin, and brassinolide’, ‘starch and sugar degradation’, and ‘redox reaction’ (Figure 3E). In the 11th leaf, 5–262 DEGs were involved in the ‘stress response to ABA, ethylene, and auxin signaling pathways’, ‘photosynthesis’, ’starch and sugar metabolism’, ‘cell apoptosis’, and ‘leaf senescence’ (Figure 3F). These processes may be crucial for the variation in tetraploid traits.

To further examine the potential biological processes affected by polyploidization, a KEGG pathway enrichment analysis was conducted on the DEGs. All significantly enriched KEGG pathways are shown in Figure 3G–I (*p* < 0.05). The DEGs were significantly enriched in the ‘flavonoid biosynthesis’, ‘plant hormone signal transduction’, and ‘circadian rhythm’ pathways in the first leaf (Figure 3G). In the fifth leaf, the DEGs were mainly enriched in the ‘galactose metabolism’, ‘starch and sucrose degradation’, and ‘plant hormone signal transduction’ pathways (Figure 3H). In the 11th leaf, the DEGs were mainly significantly enriched in the ‘plant hormone signal transduction’, ‘MAPK signal’, ‘plant–pathogen interaction’, and ’photosynthesis’ pathways (Figure 3I). It was notable that, in the GO and KEGG enrichment analyses, a large number of DEGs in the leaves at all three nodes were significantly enriched in the process of plant hormone synthesis and signal transduction.

### 2.4. Determination of Endogenous Hormones and Expression of Hormone-Signaling Genes in Diploids and Tetraploids

To examine the relationship between the phenotype and hormones after polyploidization, we measured the contents of endogenous hormones (auxin, gibberellin, JA, and ABA) in leaves of diploids and tetraploids at three nodes, and the hormone signal-related genes that encode receptors and response factors were identified by KEGG pathway enrichment analysis.

Auxin (indole-3-acetic acid, IAA) and gibberellin (GA_3_) can promote plant growth and internode elongation [25]. In the present study, the IAA and GA_3_ contents of tetraploids were significantly lower than those of diploids (Figure 4A). In the auxin signaling pathway, two auxin-responsive *SMALL AUXIN-UP RNA 36* (*SAUR36*) genes (Pop_G03G078252 and Pop_A03G050447), which promote leaf senescence, were upregulated at the three leaf nodes of tetraploids. The *AUX/IAA* transcriptional regulator family of proteins inhibits auxin signal transduction. Compared with the diploids, one *AUX/IAA* gene was upregulated in the fifth leaf, and five *AUX/IAA* genes were upregulated in the 11th leaf in the tetraploids. GH3 proteins are negative regulators in response to auxin. In tetraploids, most *GH3* genes were upregulated in the first and 11th leaves (Figure 4B; Appendix A). Moreover, in the gibberellin signaling pathway, six genes (Pop_A01G017618, Pop_A06G062163, Pop_A09G014464, Pop_G06G082035, Pop_G09G014084, and Pop_G16G025336) encoding DELLA protein were upregulated in all leaves. Two gibberellin receptor *GID1* genes (Pop_A14G000831 and Pop_G14G045139) were downregulated in the first and fifth leaves of the tetraploids (Figure 4B; Appendix A). These results suggested that the upregulation of most auxin signaling-related DEGs and a large number of DELLA proteins may contribute to the decrease in auxin and gibberellin contents in leaves of tetraploids.

JA and ABA inhibit growth and development, as well as promote plant senescence [26]. Compared with diploids, the JA and ABA contents of tetraploids were significantly increased (Figure 4A). JASMONATE ZIM-DOMAIN (*JAZ*) protein is an inhibitor of the JA signaling pathway. Here, four genes that encode JASMONATE ZIM-DOMAIN (*JAZ*) proteins were differentially expressed. Among them, one *JAZ* gene (Pop_G01G035869) was downregulated in the first leaf of tetraploids compared with that of diploids. Three *JAZ* genes (Pop_A08G086419, Pop_G08G046460, and Pop_G15G074364) were downregulated in the fifth leaf. MYC2 can regulate the biosynthesis of sesquiterpenoids and is a core regulator of the JA signal transduction pathway. In tetraploids, two *MYC2* genes (Pop_A09G015372 and Pop_G01G089277) were upregulated in the three leaves, and one *MYC2* gene (Pop_A14G044750) was downregulated in the fifth leaf (Figure 4B; Appendix A). In addition, in the ABA signaling pathway, the ABA response factors *ABF* (Pop_A09G026716 and Pop_G09G077307), *bZIP* (Pop_G09G011691), and *SnPK2* (Pop_A02G088618 and Pop_G02G065896) were upregulated in tetraploids. The ABA-encoding receptors *PYL4* (Pop_A06G089383, Pop_A16G090069, Pop_G06G051642, and Pop_G16G068642) and *PYL6* (Pop_A10G069584 and Pop_G10G048195) were not significantly differentially expressed in the first leaf, but two *PYL4* genes (Pop_A16G090069 and Pop_G06G051642) were downregulated in the fifth leaf, and all were down-regulated in the 11th leaf (Figure 4B; Appendix A).

### 2.5. Co-Expression Modules Associated with Hormone Content

The WGCNA method was used to identify co-expressed gene modules associated with phenotypic traits. A module is a cluster of genes whose expression is highly correlated. Genes in the same module have extremely high correlation coefficients with each other [27]. In the present study, the transcriptome sequencing data and four hormone indicators (GA, IAA, JA, and ABA) were used to construct the co-expression network. The genes were divided into 32 gene modules, each represented by a different color in the network (Figure 5A). The correlations between individual gene modules of each phenotype are shown in Figure 5B. The largest module (turquoise) contained 14,072 genes, whereas the smallest module (midnight blue) contained 36 genes. The gray module represented genes that could not be classified into any one module and included a total of 488 genes.

The correlations between gene modules and phenotypes were analyzed. The purple module (comprising 650 genes) showed the strongest positive correlation with IAA (*r*^2^ = 0.86, *p*-value < 0.001) and GA (*r*^2^ = 0.76, *p*-value < 0.001) (Figure 5B). A GO enrichment analysis of genes in the purple module showed that the constituent genes were mainly enriched in hormone synthesis and metabolism (GO:0010423, negative regulation of brassinosteroid biosynthetic process; GO:0090032, negative regulation of steroid hormone biosynthetic process; GO:0032351, negative regulation of hormone metabolic process; GO:0032353, negative regulation of hormone biosynthetic process; GO:0010422, regulation of brassinosteroid biosynthetic process; GO:0090030, regulation of steroid hormone biosynthetic process), cell growth and development (GO:0040007, growth; GO:0009826, unidimensional cell growth; GO:0016049, cell growth; GO:0048589, developmental growth), and other related biological processes (Appendix A). A KEGG pathway enrichment analysis of genes in the purple module showed that these genes were mainly involved in plant hormone signal transduction (kegg:04075, plant hormone signal transduction), photosynthesis (kegg:00196, photosynthesis—antenna proteins), brassinosteroid biosynthesis (kegg:00905, brassinosteroid biosynthesis), and other biological processes (Figure 5C). The cyan module (comprising 145 genes) had the strongest positive correlation with ABA (*r*^2^ = 0.77, *p*-value = 0.0002) and JA (*r*^2^ = 0.7, *p*-value = 0.001) (Figure 5B). The KEGG pathway enrichment analysis showed that these genes were mainly involved in photosynthesis (kegg:00195, photosynthesis) and plant hormone signal transduction (kegg:04075, plant hormone signal transduction) (Figure 5D). Comparison of the gene annotations in the purple and cyan modules revealed that there were complex signal networks and crosstalk between different hormone signal transduction pathways. These genes jointly regulated the growth and development of cells, and then adjusted the hormone signals, so as to regulate plant growth.

### 2.6. Association Analysis of Differentially Expressed Transcription Factors and Core Genes

Transcription factors play a crucial role in gene expression. In the present study, a total of 42 TFs, representing 19 TF families, were identified in the purple and cyan modules. Most of the TFs belonged to the MYB, GRAS, ARF, and other families (Figure 6A; Appendix A). To identify TFs significantly correlated with hormone signaling pathways, we constructed a co-expression network using the hormone signaling-related DEGs in the module (Figure 6B; Appendix A). The screening threshold for high correlation was set as the absolute value of *r* > 0.7. Larger nodes indicate stronger associations between genes. In this co-expression network, 165 pairs of corresponding relationships between 18 hormone signaling pathway genes and 28 TFs were identified. Among the hormone signaling pathway genes, *BRI1* and *SAUR*, which were relatively highly associated with TFs, were downregulated in tetraploids compared with diploids (Figure 6B; Appendix A). This result indicated that the hormone signal promoting growth was weakened. Further analysis of TFs directly associated with hormone signaling pathways in the network showed that the TFs such as MYB, G2-like, and BBX had high connectivity with hormone pathway genes in the module (log_2_FC ≥ 1 and *p*-value < 0.05). Interestingly, these TFs were also downregulated in tetraploids. These results suggested that plant hormone-related TFs were involved in the regulation of dwarfing in tetraploids, and they played important roles in the expression and transcription regulatory network.

### 2.7. qRT−PCR Validation

To verify the authenticity of the RNA-sequencing (RNA-seq) data, eight DEGs associated with IAA, GA, JA, and ABA signaling were randomly selected for real-time fluorescence quantification (Figure 7; Appendix A). The relative expression patterns of all DEGs determined by qRT−PCR were similar to those of the RNA-seq data. Thus, the results obtained by transcriptome sequencing were reliable.

## 3. Discussion

With technological progress, the advantages presented by polyploid plants and elucidation of the underlying mechanisms indicate that we will enter an era of artificially induced plant polyploidy [28]. Compared with diploids, polyploidy often leads to phenotypic changes. In the present study, the phenotypic and anatomical characteristics of diploids and tetraploids generated by chromosome doubling were compared. The plant height of the tetraploids was significantly reduced compared with that of the diploids (Figure 1A), which is similar to results reported for tetraploid *Eucalyptus* [9] and tetraploid apple [10]. The cross-sectional cell area in the leaf of tetraploid poplar 84K was larger, which was similar to previous findings for polyploid *Arabidopsis* [29] and tetraploid *Medicago sativa* [30]. Senescence was accelerated in the tetraploids (Figure 1), which was similar to previous reports for hexaploid *Populus* [12], tetraploid *Populus* [21], and tetraploid cucumber [31]. Furthermore, we also measured the endogenous hormone contents of diploids and tetraploids. The contents of IAA and GA, two hormones that promote plant growth and development [25], were significantly increased in tetraploid poplar 84K. In contrast, JA and ABA, which inhibit plant growth [26], were significantly reduced in tetraploids (Figure 4A). These results are consistent with previous studies of the dwarf shengyin bamboo (*Phyllostachys edulis* f. *tubaeformis*) [32], tetraploid apple [10], and dwarf mutants of tartary buckwheat [33].

Previous studies have shown that polyploidy results in changes in gene expression levels [34]. To evaluate the changes at the molecular level after polyploidization of poplar 84K, the current study used the genome sequence of poplar 84K [35] as a reference and RNA-seq technology to compare the transcriptome of leaves in tetraploids and diploids. Through comparative analysis, 7917 DEGs were detected (Figure 2A). Among the differentially expressed genes in tetraploids and diploids studied by predecessors, phytohormone-related genes comprised an important assemblage [19,20]. Therefore, GO and KEGG enrichment analyses of the DEGs revealed that a large number of DEGs were involved in hormone signaling pathways (Figure 3G–I; Appendix A). To explore the molecular mechanism via which hormones affect plant growth and to determine the relationship between DEGs and the dwarf phenotype of tetraploid poplar 84K, we focused attention on the crucial genes involved in four phytohormone signaling pathways.

Auxin is an important growth regulator and plays an important role in plant growth and development [36]. *AUX/IAA*, *GH3*, and *SAUR* are considered to be early and major auxin-response genes [37]. Among these genes, the *AUX/IAA* family plays a crucial role in inhibiting the expression level of genes activated by auxin response factors (ARFs) [38,39]. In the current study, four *AUX/IAA4* genes (Pop_A05G001751, Pop_G02G023725, Pop_G05G057440, and Pop_G05G072154) were significantly upregulated in the 11th leaf of tetraploids, and two *AUX/IAA13* genes (Pop_A10G047408 and Pop_G10G064704) were significantly upregulated in the fifth and 11th leaves, respectively. The *GH3* gene family is involved in hormonal homeostasis by binding to free forms of IAA and JA [40,41]. Overexpression of *GH3.1* and *GH3.6* can significantly reduce the auxin content and lead to dwarfism in rice [42,43]. Compared with diploids, three *GH3.1* genes and one *GH3.6* gene were significantly upregulated in the first and 11th leaves of tetraploids. In addition, GH3.5 and LAX3 play positive roles in regulating the auxin signaling pathway [44,45]. In tetraploids, one *GH3.5* gene (Pop_A19G055012) was downregulated in the leaves at all three nodes, and one *LAX3* gene (Pop_A02G088716) was downregulated in the fifth leaf. *SAUR36* is a positive regulator of leaf senescence and mediates auxin-induced leaf senescence in *Arabidopsis* [46,47]. In the present study, two *SAUR36* genes were significantly upregulated in tetraploids (Figure 4B; Appendix A). Thus, the changes in expression of these crucial genes involved in the auxin signaling pathway were integral to the reduction in auxin content of the tetraploids. The decrease in IAA content may be one reason for the slow growth of the tetraploid plants.

Gibberellin plays an important role in regulating plant growth and development [48]. Gibberellin is a vital factor in the control of plant height. For example, dwarfism results if the genes involved in GA synthesis or signal transduction are mutated in sponge gourd (*Luffa acutangula*) [49]. Increasing evidence indicates that GA promotes nutritional and reproductive growth by triggering the degradation of DELLA proteins, which are the main inhibitors of GA signal transduction [50,51]. In the present study, six genes encoding DELLA proteins were significantly upregulated in all three sampled leaves of tetraploids. The upregulation of DELLA proteins will lead to the reduction in GA content. In addition, the gibberellin receptor GID1 binds to bioactive gibberellins (e.g., GA_1_, GA_3_, and GA_4_) to degrade DELLA proteins, thereby relieving DELLA-mediated growth inhibition [52,53]. Compared with diploids, two *GID1* genes (Pop_A14G000831 and Pop_G14G045139) were downregulated in the first and fifth leaves of tetraploids (Figure 4A; Appendix A). Therefore, GID1 expression was downregulated, leading to DELLA protein accumulation, thus reducing the GA content and resulting in dwarfism of the plants.

Jasmonic acid is a lipid-derived plant hormone that is essential for plant defense and plant development, and it is an important endogenous signal that activates the expression of senescence-related genes and induces leaf senescence [54,55]. A critical element in JA signaling is *JAZ* proteins, which are inhibitors of JA signaling [56]. In addition, *MYC2* acts upstream of the JA signaling pathway and functions as the regulatory center for plant hormone signal transduction by integrating various endogenous and exogenous signals that affect plant growth and development [57]. Previous studies have shown that *MYC2* plays an active role in JA-mediated leaf senescence [58]. In the current study, four genes encoding *JAZ* proteins were downregulated in the first and fifth leaves of tetraploids, and two genes encoding *MYC2* were upregulated in the three leaves sampled (Figure 4A; Appendix A). Therefore, the downregulated expression of *JAZ* proteins in the tetraploids reduced the repression of *MYC2* and activated JA to induce leaf senescence.

Abscisic acid is considered to be a growth inhibitor, and an increase in the ABA content inhibits germination and causes growth retardation [59]. *SnPK2* is a positive regulator of ABA signaling in *Arabidopsis thaliana* [60]. The released SNF1-RELATED PROTEIN KINASE 2 (*SnRK2*) is then activated by autophosphorylation or phosphorylation by other kinases, and the activated *SnRK2* can phosphorylate downstream proteins or transcription factors, including ABA response element binding factors (*AREB/ABF*) and *bZIP* TFs [61]. In the present investigation, two *SnPK2* genes, two *ABF3* genes, and one *bZIP* gene were upregulated in tetraploids, and the number of upregulated genes was greatest in the 11th leaf (Figure 4A; Appendix A). The results showed that the upregulated expression of *SnPK2*, *ABF3*, and *bZIP* genes in the ABA signaling pathway led to accumulation of ABA and, thus, accelerated aging.

The dwarf phenotype of plants is not only affected by the synergistic and antagonistic effects of hormones, but also by the regulatory effects of TFs. The role of TFs is to transform external signals into intracellular signals, thereby stimulating specific hormone signaling pathways and gene expression to maintain growth [62]. Here, the WGCNA co-expression network was used to explore the regulatory relationship between TFs and pathway genes. It was observed that *MYB*, *G2-like*, and *BBX* genes were downregulated in all three sampled leaves, and *BRI1*, *SAUR*, and other pathway genes strongly correlated with TFs were downregulated (Figure 6; Appendix A). Plant *MYB* proteins have been reported to play a role in hormonal responses during seed development and germination [63,64]. *MYB33* mediates GA signaling in plant growth and is a positive regulator of GA signal transduction [65]. *MYB61* activates the expression of *GA3ox1* and *GA3ox2* genes, which in turn may lead to increased GA_4_ content in *Artemisia annua* and *Arabidopsis thaliana* [66]. In the present study, one *MYB33* gene and two *MYB61* genes were all downregulated in tetraploids, which may have inhibited GA. In addition, *G2-like* (*GLK*) TFs play important roles in regulating plant aging and hormone contents [67,68]. Overexpression of *GLK* in maize and tomato promotes chloroplast development and the expression of a photosynthesis-related gene [69]. In tobacco, *GLK* inhibits ABA signaling, and the expression of *GLK* gradually decreases with the aging of leaves [70]. A *GLK* gene was downregulated in the fifth and 11th leaves of tetraploid poplar 84K, and it might participate in ABA signaling as a negative regulator and accelerate leaf senescence. *BBX* proteins play an important role in light-regulated signaling pathways and plant hormone-mediated development. *BBX* negatively regulates the expression of *ABI*, a positive regulator of ABA signaling [71]. *BBX18* is involved in the GA signaling pathway and increases bioactive GA contents to promote growth by enhancing the expression of the GA metabolic genes *GA3ox1* and *GA20ox1* [72]. A *BBX18* gene was downregulated in the 11th leaf of tetraploid poplar 84K, which may have led to the reduction in GA content observed in this study. These changes may have reduced the accumulation of IAA and GA, increased the accumulation of JA and ABA, and inhibited the growth and development of tetraploid poplar 84K.

## 4. Materials and Methods

### 4.1. Plant Materials and Growth Conditions

Diploid (2*n* = 2*x* = 38) and tetraploid (2*n* = 4*x* = 96) individuals of poplar 84K were used as materials in this study. Tetraploids were obtained by colchicine-induced doubling of the somatic chromosomes of diploids [24]. Rooting culture was conducted in half-strength Murashige–Skoog medium supplemented with 0.02 mg/mL 1-naphthaleneacetic acid, 0.05 mg/mL indole-3-butyric acid, 30 g/L sucrose, and 6.5 g/L agar (the culture medium and conditions of all plants were consistent). After culture for approximately 30 days, 15 diploid and 15 tetraploid plants of uniform growth were selected and transplanted into 22 cm × 22 cm plastic pots containing nutrient soil, perlite, and vermiculite (2:1:1, *v*/*v*/*v*). All plants were cultivated in the greenhouse of Beijing Forestry University at 25 °C with a 16 h light/8 h dark photoperiod. Healthy plants of uniform growth were selected for following experiments.

### 4.2. Measurement of Phenotypic Traits and Anatomical Observations

The plant height (*H*) of 15 diploid and 15 tetraploid plants of poplar 84K was measured with a steel tape once per month for a total of 5 months. The fully expanded fifth leaf (including the main leaf vein) and the stem internode between the fourth and fifth leaves (numbered from the shoot tip) were sampled from the tetraploids and diploids. The samples were fixed in FAA fixative (0.25 mL of 38% formaldehyde, 0.25 mL of glacial acetic acid, and 4.5 mL of 50% ethanol) for more than 24 h. The fixed samples were dehydrated sequentially in an ethanol gradient series (50%, 70%, 95%, and 100%), then infiltrated with 100% xylene for 30 min, and embedded in paraffin at 60 °C for 2 h. After solidification, the embedded sample was trimmed to a suitably sized trapezoid block, and sections of thickness ~8 μm were cut with a microtome. After dewaxing, the sections were stained with 1% safranin O and 0.1% fast green. All sections were observed and photographed with an Olympus BX51 microscope.

### 4.3. Total RNA Extraction and Transcriptome Sequencing

The materials to be sampled were determined by analyzing the photosynthetic rate of diploids and tetraploids (Appendix A). Therefore, the first, fifth, and 11th leaves were sampled simultaneously when the plants were 3 months old. Three replicates were randomly selected for each ploidy. The sampled materials were immediately frozen in liquid nitrogen and stored at −80 °C for subsequent sample sequencing and measurement of physiological indicators. The TRIzol Reagent Kit (TIANGEN Biotech Co., Ltd., Beijing, China) was used to extract RNA from the samples in accordance with the manufacturer’s instructions. A NanoDrop 2000 spectrophotometer (Thermo Fisher Scientific Inc., Wilmington, DE, USA) was used to determine the RNA quality. The integrity of the RNA was verified by electrophoresis in 1.5% agarose gel. After the RNA quality was confirmed, the extracted RNA was reverse-transcribed, and cDNA libraries constructed with the FastKing RT Kit (with gDNase). The libraries were sequenced on an Illumina HiSeq platform at the Hangzhou Lianchuan Biological Technology Co., Ltd. (Beijing, China).

After filtering all low-quality reads, clean reads were used to calculate the base quality score Q20 and Q30 values [73]. The resulting clean data were aligned to the poplar 84K reference genome of Qiu et al. [35] (http://ftp.cngb.org/pub/CNSA/data1/CNP0000339/CNS0047055/CNA0003521/) (accessed on 24 October 2021) with HISAT2 (https://daehwankimlab.github.io/hisat2) (accessed on 24 October 2021). StringTie software (http://ccb.jhu.edu/software/stringtie) (accessed on 24 October 2021) was used for initial assembly of genes and transcripts, and the initial assembly results for all samples were combined. Lastly, mapping of the transcripts to the reference genome was conducted using GffCompare (http://ccb.jhu.edu/software/stringtie/gffcompare.shtml) (accessed on 24 October 2021), and the final assembly annotations were obtained.

### 4.4. Enrichment Analysis of DEGs

The transcript abundance was calculated as reads per kilobase per million mapped reads (RPKM). The edgeR software (https://bioconductor.org/packages/release/bioc/html/edgeR.html) (accessed on 24 October 2021) was used for differential expression analysis between samples [74]. Genes with fold difference multiples of |log_2_(FC)| > 1 and *p*-value < 0.05 were defined as DEGs. In addition, Gene Ontology (GO) and Kyoto Encyclopedia of Genes and Genomes (KEGG) pathway enrichment analyses were performed. Using the GO database (Gene Ontology, 2021), the DEGs were annotated using the GO online tools with terms classified into three categories: molecular function, biological process, and cell composition. Using the KEGG database (http://www.kegg.jp/) (accessed on 17 June 2022), the DEGs were annotated with signaling pathways and metabolic pathways.

### 4.5. Determination of Endogenous Hormone Contens

Briefly, 0.5 g of the first, fifth, and 11th leaves stored at −80 °C were used for the determination of hormone content. Contents of the plant hormones IAA, GA_3_, ABA, and JA were determined using an enzyme linked immunosorbent assay in accordance with a previously described method [75]. Each treatment had three biological repetitions and three technical repetitions.

### 4.6. Division of Co-Expression Modules and Visualization of Gene Expression

The gene co-expression network was constructed using the WGCNA package (version 4.0.2) for R software. To maintain gene connectivity and place greater weight on the strongest correlation, a soft threshold for the correlation matrix (β) was selected. The hierarchical clustering dendrogram of the TOM matrix was constructed using the average distance with a minimum size threshold of 30 and the merge cut height of 0.25. Genes with similar expression patterns were clustered and divided into unified modules, and a co-expression network was constructed. The correlations among gene modules were analyzed using Pearson correlation analysis, and the expression patterns of modules and samples were calculated. The FPKM values of the genes were used to map the network of pathways, genes, and TFs. The screening threshold was |*R*| ≥ 0.7. Positive values indicated a positive correlation, and negative values indicated a negative correlation. The PlantTFDB database (http://planttfdb.gao-lab.org/) (accessed on 17 June 2022) was used to identify genes encoding TFs among the DEGs. The gene co-expression networks were visualized using Cytoscape 3.8.2, and the same software was used to calculate the gene connectivity. The size of the nodes was positively correlated with the degree of gene connection.

### 4.7. qRT−PCR Verification of Differentially Expressed Genes

RNA was extracted from the leaves of diploid and tetraploid individuals. After determining the RNA concentration and quality, the first-strand cDNA was reverse-transcribed using the FastKing RT Kit (TIANGEN Biotech Co., Ltd., Beijing, China). Nine DEGs were selected for verification by qRT-PCR analysis, among which the constitutively expressed *Actin* gene was used as the internal reference gene (GenBank accession number: EF145577). Real-time PCR amplifications were performed on the Applied Biosystems 7500 Fast Instrument (AB Ltd., Lincoln, NE, USA) using the 2 × SYBR^®^ Green qPCR Mix Kit (Aidlab, China). Gene-specific primers for the nine DEGs were designed using the Primer3 Plus online tool (http://www.primer3plus.com/) (accessed on 17 June 2022). The primer sequences are listed in Appendix A. The RT-qPCR protocol consisted of 40 cycles, comprising initial denaturation at 95 °C for 5 min, denaturation at 95 °C for 10 s, annealing at 60 °C for 30 s, and extension at 72 °C for 15 s. The relative expression level of genes was calculated using the 2^−∆∆Ct^ method [76]. All reactions comprised at least three biological replicates and three technical replicates.

### 4.8. Statistical Analysis

The experimental data were statistically analyzed using SPSS 20.0 (IBM Corporation Inc., Armonk, NY, USA). The results were expressed as the mean ± standard deviation. The significance of the difference among means was determined by Student’s *t*-test at *p* < 0.05, *p* < 0.01, and *p* < 0.001. The cell area, length, and width for the leaves and stems were determined with ImageJ. TBtools was used to draw the heatmaps [77].

## 5. Conclusions

The molecular mechanism of trait variation in plant polyploids is highly complex. Hormone synthesis, signal transduction, and gene regulation are all associated with phenotypic changes. In this study, compared with the diploids, most DEGs positively correlated with growth in the tetraploids were down-regulated, consistent with IAA and GA contents in the leaves, and most DEGs negatively correlated with growth were upregulated, in accordance with JA and ABA contents in the leaves. The changes in endogenous hormone contents were consistent with transcriptomic changes. In addition, co-expression network analysis showed that TFs may also play important regulatory roles that together lead to tetraploid dwarfing. The present results provide a theoretical basis for future research on the mechanism of plant trait variation at the molecular level in polyploid woody plants.

## Figures and Tables

**Figure 1 ijms-23-09762-f001:**
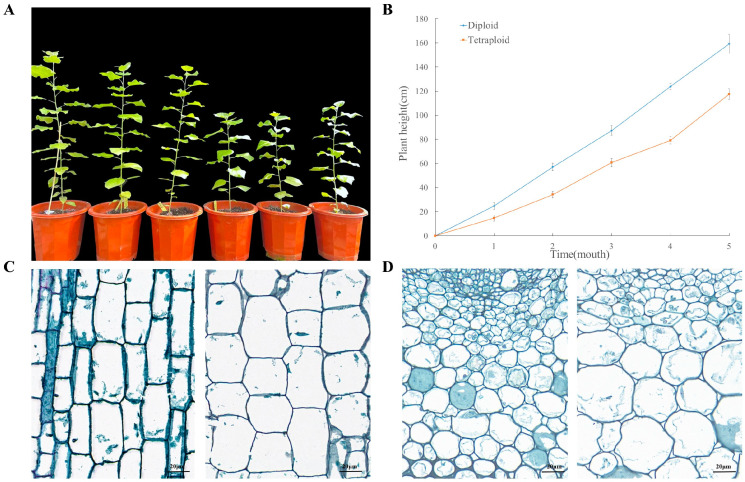
Comparison of plant height and anatomical characteristics of diploid and tetraploid poplar 84K. (**A**) Phenotype of tetraploids and diploids after 3 months. (**B**) Average growth rates (height) between tetraploids and diploids during time. (**C**) Longitudinal sections of the diploid (left) and tetraploid (right) stem internodes. (**D**) Transverse sections of the diploid (left) and tetraploid (right) leaves.

**Figure 2 ijms-23-09762-f002:**
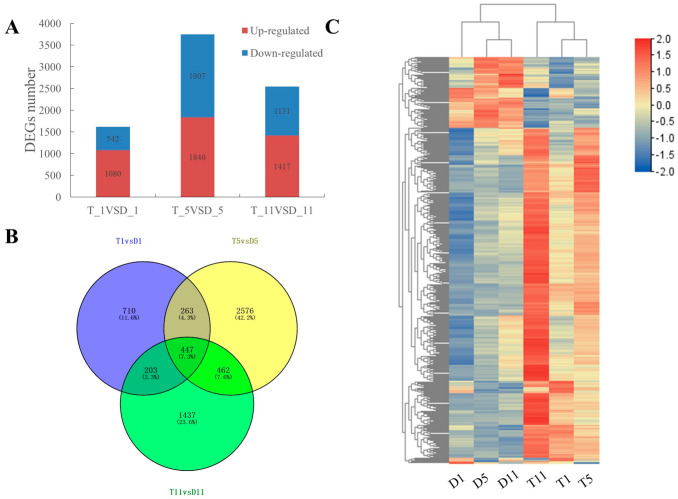
Analysis of transcriptome sequencing data for leaves of diploid and tetraploid poplar 84K. (**A**) Number of upregulated and downregulated differentially expressed genes (DEGs) in the first, fifth, and 11th leaves of tetraploids. (**B**) Statistics for DEGs in leaves of diploids and tetraploids at three nodes. (**C**) Cluster analysis of the samples based on RNA-sequencing data.

**Figure 3 ijms-23-09762-f003:**
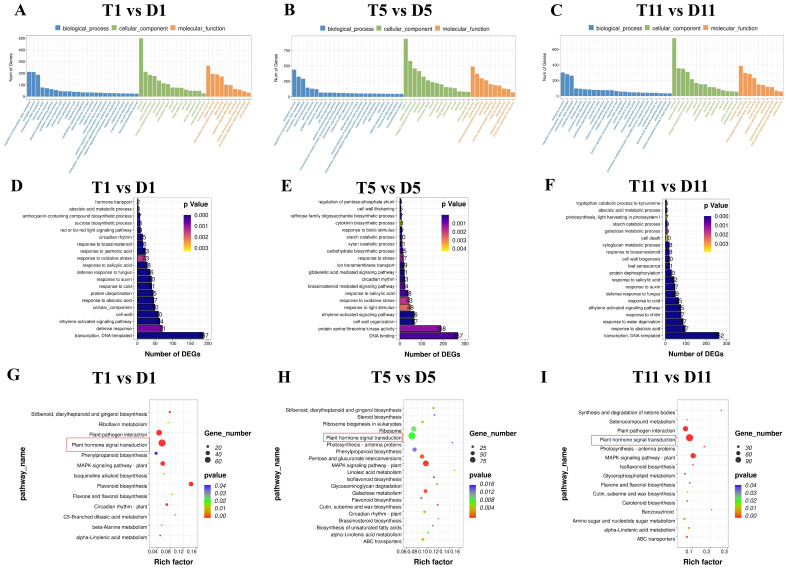
Functional enrichment analysis of differentially expressed genes (DEGs) between diploid and tetraploid poplar 84K. (**A**–**C**) Gene ontology (GO) analysis of DEGs in the first leaf (**A**), fifth leaf (**B**), and 11th leaf (**C**). (**D**–**F**) The 20 most enriched biological process GO terms in the first leaf (**D**), fifth leaf (**E**), and 11th leaf (**F**). (**G**–**I**) Kyoto Encyclopedia of Genes and Genomes (KEGG) analysis of enriched pathways among DEGs in the first leaf (**G**), fifth leaf (**H**), and 11th leaf (**I**).

**Figure 4 ijms-23-09762-f004:**
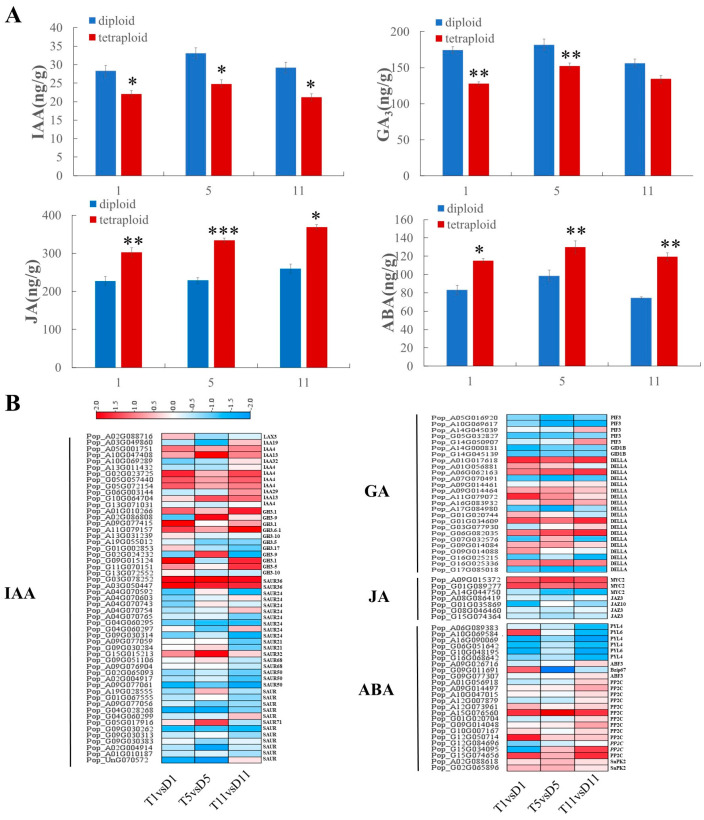
Differentially expressed genes (DEGs) of transcription-level hormone pathways leading to changes in hormone contents in tetraploid poplar 84K. (**A**) Comparison of endogenous hormone contents in leaves at three nodes of diploids and tetraploids; * *p* < 0.05, ** *p* < 0.01, *** *p* < 0.001. (**B**) DEGs associated with auxin (IAA), gibberellin (GA), abscisic acid (ABA), and jasmonic acid (JA) in tetraploids and heatmap of the log_2_ fold change. Red and blue colors indicate up- and downregulation, respectively.

**Figure 5 ijms-23-09762-f005:**
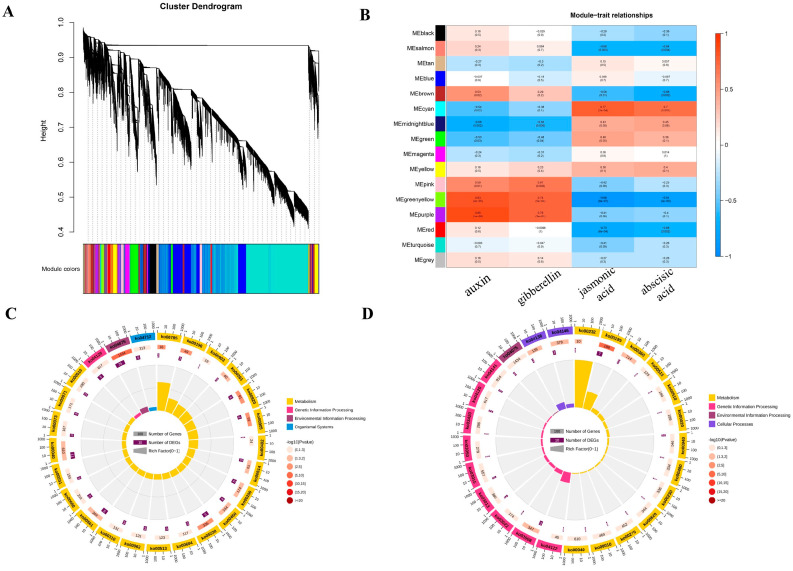
Weighted gene co-expression network analysis (WGCNA) of RNA-sequencing data from diploid and tetraploid poplar 84K. (**A**) Hierarchical clustering dendrogram of co-expression modules identified by WGCNA. Every branch on the tree represents one gene. (**B**) Association analysis of gene co-expression network modules with hormone contents. The horizontal axis represents different samples, and the vertical axis represents the eigenvectors of each module. The red grid represents a positive correlation between the physiological trait and the module. (**C**) KEGG pathway enrichment analysis of genes in the purple module. (**D**) KEGG pathway enrichment analysis of genes in the cyan module.

**Figure 6 ijms-23-09762-f006:**
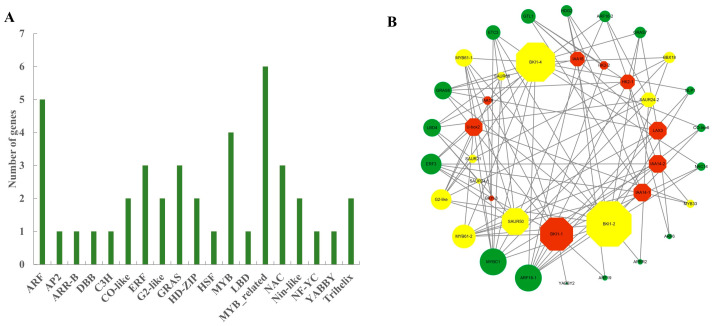
Analysis of differentially expressed transcription factors (TFs) in target gene co-expression network modules. (**A**) Distribution of differentially expressed TFs in the purple and cyan modules. (**B**) Co-expression network between pathway genes (hormone) and TFs. The green circular nodes represent TFs. The red octagonal nodes represent the pathway genes of the plant hormone. Among these nodes, yellow nodes represent significant differences (*p*-value < 0.05). The node size is positively correlated with the gene degree.

**Figure 7 ijms-23-09762-f007:**
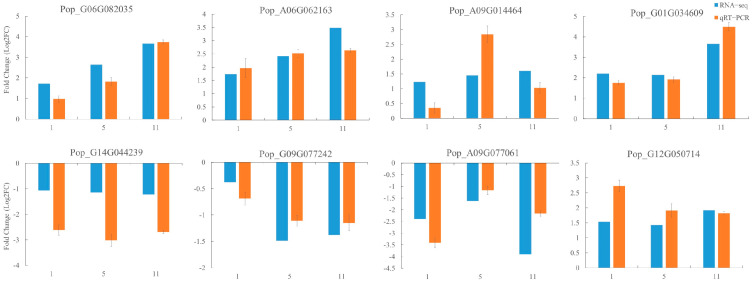
qRT−PCR validation of the transcriptome data.

## Data Availability

The datasets supporting the conclusions of this article are included within the article and its additional files.

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
