# Peer review of "Morphological, Transcriptome, and Hormone Analysis of Dwarfism in Tetraploids of Populus alba × P. glandulosa"

_ijms, 2022, doi:10.3390/ijms23179762_

Round 1
Reviewer 1 Report
This manuscript entitled “Morphological, transcriptome, and hormone analysis of dwarf-2 ism in tetraploids of Populus alba × P. glandulosa” by Yongyu Ren Shuwen Zhang , Tingting Xu and Xiangyang Kang provides insight into molecular differences at transcript and hormonal levels between diploid and tetraploid genotypes of Populus. The authors show lower indole-3-acetic acid (IAA) and gibberellin (GA3) contents and higher jasmonic acid 16 (JA) and abscisic acid (ABA) contents in leaves of the tetraploids with reduced growth. The transcriptome sequencing data analyses provide the backgrounds for hormonal status of the analyzed leaf tissues. In tetraploids, most GH3 genes and DELLA protein genes were up-regulated. The two MYC2 from the JA signal transduction pathway were up-regulated in the three leaves. This work describes correlation analyses between gene modules and phenotypes. Considering the presented data and quality of the experiments performed I can support the publication after additional explanations.
The author may consider the followings:
1.The reduced growth rate of tetraploids is a key feature. Therefore, we need explanation for the use of leaf explants for the present study. Analysis of shoot meristem tissues could provide more reliable information in relation to plant stem growth.
2. The plant growth rate is dependent on the age of plants?
3. Pre-culture in auxin containing medium how long can influence plant behavior?
4. The transcriptome analysis presented in this work is novel and original. However, the growth and hormone infromations are confirmative. See: Dudits et al. ( 2016) Response of Organ Structure and Physiology to Autotetraploidization in Early Development of Energy Willow Salix viminalis. Plant Physiology_, March 2016, Vol. 170, pp. 1504–1523. Increase in jasmonic acid an GA4 was already reported. In addition the tetraploid showed elevated salicylic acid levels.
5.List of references could be focused and reduced.
As a conclusion recommend the publication with some improvement.
Author Response
请参阅附件。

Reviewer 2 Report
The research comprises a comprehensive investigation into factors that impact genetic, hormonal regulation and morphological differences between diploid and tetraploid Populus taxa. Several lines of evidence provide convincing data that support regulatory factors. The paper is comprehensive with an extensive bibliography. The conclusions are appropriate, based on good statistical analyses.
L26: One minor suggestion; replace "lays" with provides a foundation..
